# Prosociality predicts labor market success around the world

Fabian Kosse [1✉] & Michela M. Tincani[2]

A large literature points to the importance of prosociality for the well-being of societies and individuals. However, most of this work is based on observations from western, educated, industrialized, rich, and democratic (WEIRD) societies, questioning the generalizability of these findings. Here we present a global investigation of the relation between prosociality and labor market success. Our analysis uses experimentally validated measures of prosociality and is based on about 80,000 individuals in 76 representative country samples. We show a sizable and robust positive relation between prosociality and labor market success around the world that does not systematically differ across continents or by countries' economic development. These findings generalize the positive relation between prosociality and labor market success to a wide geographical context.

[1] Ludwig-Maximilians-Universität München, Geschwister-Scholl-Platz 1, 80539 Munich, Germany. [2] University College London, Gower Street, London WC1E 6BT, UK. ✉email: fabian.kosse@econ.lmu.de

Elements of prosociality, such as reciprocity, altruism, and trust, are fundamental drivers of human interactions, and affect a wide range of economic decisions. A large literature points to their importance for the functioning of societies and markets[1–5]. While trust has been shown to be a driver of economic exchange[6], reciprocity can act as a contract enforcement device and therefore lead to more efficient market outcomes[7]. More recent evidence also points to the benefits of prosociality and social skills for individuals' life-outcomes[8–10], which is in-line with findings from a large body of literature on the returns to non-cognitive or socio-emotional skills[11–14]. To describe the potential underlying processes, Deming[9] presents a model, which suggests that social skills reduce coordination costs, allowing workers to specialize and work together more efficiently. However, most of the evidence on the importance of prosociality is based on observations from western, educated, industrialized, rich, and democratic (WEIRD) societies, which calls the generalizability of these results into question[15]. Differences among societies might, e.g., evolutionarily arise from heterogeneities in market integration, community size, and religious institutions (see Henrich et al.[16] for a detailed theoretical and empirical discussion).

We contribute to the literature by providing correlational evidence on the link between prosociality and labor market success in 76 representative country samples, which, taken together, represent about 90% of the world population. Our results underline the generality of the positive relationship between prosociality and labor market outcomes. For the analyses we use data from the Global Preference Survey (GPS)[17], which includes experimentally validated measures of prosociality and covers about 80,000 individual observations.

## Results

All regressions include subnational region fixed effects. Standard errors are clustered at the country level and observations are weighted using sampling weights to achieve (ex post) representativeness[17]. All analyses were conducted using Stata 14.2. We begin our analyses by regressing log household income on the prosociality measure, which is standardized to have mean zero and standard deviation one in the joint sample of all countries.

The results, shown in Table 1 Panel A, indicate that prosociality positively predicts household income. We label this link, which is not necessarily causal, as the income premium of prosociality. Without further controls, an increase in prosociality by one standard deviation predicts an ~8% higher household income. As labor market outcomes and prosociality might both be determined by socio-demographics, we control for gender and age in column 2. However, these controls do not affect the relation between prosociality and earnings. Controlling for cognitive ability moderately reduces the coefficient of prosociality (column 3). While controlling for cognitive ability is common in the literature[13], the interpretation of this effect is ambiguous due to a bad control problem[18]. Controlling for cognitive ability would lead to an underestimation of the effect of prosociality if cognitive ability itself were a function of prosociality. This may be the case if both characteristics develop jointly and interactively[19,20]. However, it is reassuring that prosociality predicts income above and beyond cognitive ability. Finally, in column 4, in order to avoid potential biases due to intrahousehold division of labor, we restrict the sample to individuals who do not live with a partner. The results indicate a very similar relation between prosociality and income for singles and families.

In the next step of the analysis, we focus on the individual-level measures of labor market success and regress binary indicators of underemployment and unemployment on prosociality. The results regarding underemployment, shown in Table 1 Panel B, indicate that prosociality is negatively related to the probability of being underemployed. Relative to the unconditional probability of being underemployed of 21.3%, an increase of one standard deviation in prosociality is related to a 6.3% (~1.3 percentage points) lower probability of being underemployed. This result is

**Table 1 Prosociality predicts labor market success.**

|  | (1) | (2) | (3) | (4) |
|---|---|---|---|---|
| **Panel A** | **Log household income** | | | |
| Prosociality (standardized) | 0.079*** | 0.078*** | 0.060*** | 0.058*** |
|  | (0.009) | (0.009) | (0.009) | (0.012) |
| Controlling for gender, age, age² | No | Yes | Yes | Yes |
| Controlling for cognitive ability | No | No | Yes | Yes |
| Sample restriction: not in partnership | No | No | No | Yes |
| Observations | 77,522 | 77,522 | 77,522 | 32,074 |
| **Panel B** | **Underemployed (0/1)** | | | |
| Prosociality (standardized) | −0.013*** | −0.013*** | −0.011*** | −0.013** |
|  | (0.004) | (0.004) | (0.004) | (0.006) |
| Controlling for gender, age, age² | No | Yes | Yes | Yes |
| Controlling for cognitive ability | No | No | Yes | Yes |
| Sample restriction: not in partnership | No | No | No | Yes |
| Observations | 45,677 | 45,677 | 45,677 | 17,314 |
| **Panel C** | **Unemployed (0/1)** | | | |
| Prosociality (standardized) | −0.009*** | −0.009*** | −0.008** | −0.012* |
|  | (0.003) | (0.003) | (0.003) | (0.006) |
| Controlling for gender, age, age² | No | Yes | Yes | Yes |
| Controlling for cognitive ability | No | No | Yes | Yes |
| Sample restriction: not in partnership | No | No | No | Yes |
| Observations | 45,677 | 45,677 | 45,677 | 17,314 |

Coefficients are OLS estimates, standard errors (clustered at country level) are displayed in parentheses, observations are weighted by the sampling weights provided by Gallup to achieve (ex post) representativeness. All regressions include subnational region fixed effects. Cognitive ability is proxied by self-reported maths skills[17]. Coefficients of the control variables are shown in Supplementary Table 1. Supplementary Table 2 displays correlations among all variables. Data source: GPS and Gallup World Poll (76 countries). Significance levels regarding two-sided $t$-tests: $*p < 0.1$, $**p < 0.05$, $***p < 0.01$.

robust across all four specifications (column 1–4). Note that underemployment is a subjective measure, as it relies on subjective beliefs on the appropriate capacity of work, which could directly be affected by prosociality. However, the measure of unemployment is more objective and the results regarding unemployment, shown in Table 1 Panel C, mirror the pattern regarding underemployment. Relative to the unconditional probability of being unemployed of 11.7%, an increase of one standard deviation in prosociality is related to a 7.8% (~0.9 percentage points) lower probability of being unemployed.

In Table 2 we further check for potential within-country nonlinear relations between labor market outcomes and prosociality by regressing labor market success on prosociality quartile dummies. In-line with the previous analysis, the results indicate more labor market success for higher quartiles and suggest positive marginal effects of prosociality for most parts of the distribution. However, the effects are relatively stronger at the bottom and middle parts of the distribution and relatively smaller at the top, which suggests a concave relationship between labor market success and prosociality. In Table 3 we explore the predictive power of the individual facets of prosociality: altruism, positive reciprocity, and trust. The estimates indicate that all three facets individually predict labor market success, with positive reciprocity showing the highest wage premium, and trust the lowest.

In the last step, we analyze heterogeneity across countries. In Fig. 1 we show country-specific estimates of the income premium of prosociality. The displayed coefficients are by-country estimates of the model shown in Table 1, Panel A, column 2. In-line with the

above presented results of the pooled sample, the figure indicates that the income premia of prosociality are significantly positive for the large majority of countries. Notable exceptions are Canada and Pakistan, for which the premia are significantly negative.

To explore the heterogeneity among country-specific income premia, we further regress the estimated income premia on continent dummies. A Wald test does not reject the null hypothesis that all dummies are jointly zero ($p = 0.520$, $N = 76$), which indicates that income premia do not significantly differ across continents. Similarly, we find no statistically significant difference in income premia between WEIRD countries, proxied by North America, Europe and Australia, and non-WEIRD countries, proxied by Asia, Africa and South America ($p = 0.711$, two-sided $t$-test, $N = 76$). Moreover, we also explore the relation between countries' income premia of prosociality, countries' average level of prosociality and countries' economic development. Figure 2 displays the relation between income premia and average level of prosociality and indicates no systematic relationship (Spearman's $\rho = -0.091$, $p = 0.436$, $N = 76$). Figure 3 displays the relation between income premia and log GDP per capita and also indicates no systematic relationship (Spearman's $\rho = -0.065$, $p = 0.579$, $N = 76$).

## Discussion

We contribute to the debate regarding the generalizability of results in the social sciences[15] by showing that around the globe, prosociality is generally positively related to labor market success. The finding robustly holds in a pooled sample, which represents about 90% of the world population and the pattern also holds for the majority of individual countries. Like most other related studies[13], ours has no ambition to tease out causality. The cross-sectional nature of our data does not allow us to explore reverse causality or omitted variables.

Between-country analyses show that income premia of prosociality are not systematically different across continents and are not related to countries' average level of prosociality or countries' economic development. This further underlines the generality of the positive relationship between prosociality and labor market outcomes, and suggests that recent models[9] of labor markets that include prosociality and social skills are likely to have a broad scope. However, our findings also reveal a need for future research on the determinants of the between-country heterogeneity.

Our findings are also relevant from a policy perspective: combining evidence on the labor market relevance of prosociality with recent evidence on its malleability[10,21] suggests prosociality as a promising target of policy interventions.

**Table 2 Analyses of nonlinear relations between labor market success and prosociality.**

|  | (1) Log HH income | (2) Underemployed | (3) Unemployed |
|---|---|---|---|
| Base: Prosociality 1st quarter |  |  |  |
| Prosociality in 2nd quarter (dummy) | 0.088*** (0.019) | −0.019* (0.010) | −0.011* (0.006) |
| Prosociality in 3rd quarter (dummy) | 0.155*** (0.020) | −0.038*** (0.010) | −0.021** (0.008) |
| Prosociality in 4th quarter (dummy) | 0.188*** (0.024) | −0.033*** (0.012) | −0.021** (0.008) |
| Observations | 77,522 | 45,677 | 45,677 |

Coefficients are OLS estimates, standard errors (clustered at country level) are displayed in parentheses, observations are weighted by the sampling weights provided by Gallup to achieve (ex post) representativeness. All regressions include subnational region fixed effects. HH means household. Base category are individuals with prosociality in the bottom 25% of the global distribution. Prosociality in 2nd quarter is a dummy variable indicating whether an individual's prosociality lies above the bottom 25% and below the median. Prosociality in 3rd quarter is a dummy variable indicating whether an individual's prosociality lies above the median and below the top 25%. Prosociality in 4th quarter is a dummy variable indicating whether an individual's prosociality lies in the top 25%. Data source: GPS and Gallup World Poll (76 countries). Significance levels regarding two-sided $t$-tests: *$p < 0.1$, **$p < 0.05$, ***$p < 0.01$.

**Table 3 Altruism, reciprocity and trust predicts labor market success.**

|  | (1) | (2) | (3) |
|---|---|---|---|
|  | Log household income |  |  |
| Altruism (standardized) | 0.057*** (0.008) |  |  |
| Positive reciprocity (std) |  | 0.074*** (0.007) |  |
| Trust (standardized) |  |  | 0.016** (0.007) |
| Observations | 77,522 | 77,522 | 77,522 |

Coefficients are OLS estimates, standard errors (clustered at country level) are displayed in parentheses, observations are weighted by the sampling weights provided by Gallup to achieve (ex post) representativeness. All regressions include subnational region fixed effects and controls for age and gender (see Table 1, column 2). Data source: GPS and Gallup World Poll (76 countries). Significance levels regarding two-sided $t$-tests: *$p < 0.1$, **$p < 0.05$, ***$p < 0.01$.

## Methods

**Sample.** Our analyses are based on the GPS, which was collected as part of the Gallup World Poll 2012 and covers representative samples of 76 countries, including 14 countries from Africa, 22 from Asia, 8 from South America, 7 from North America, 24 from Europe, and Australia. The median sample size is 1000 participants per country. The samples include one randomly selected respondent per household. For details regarding sampling schemes, data collection protocols, translation schemes, and pretests, see Falk et al.[17] and Falk and Hermle[22].

**Labor market outcomes.** Although the Gallup World Poll does not include information on individuals' earnings, it does provides three internationally comparable measures of labor market success: (1) household-level annual income in international dollars (purchasing power parity), (2) an individual-level binary measure of underemployment that indicates whether an individual is working less than desired (including self-employment), and (3) an individual-level binary measure of unemployment indicating if an individual is not employed (including self-employment) but actively looking for a job. (2) and (3) are only defined for individuals that are part of the labor force and exclude full-time students, retired and disabled individuals, and homemakers. In the analyses we explore the relation of prosociality with all three measures of labor market success.

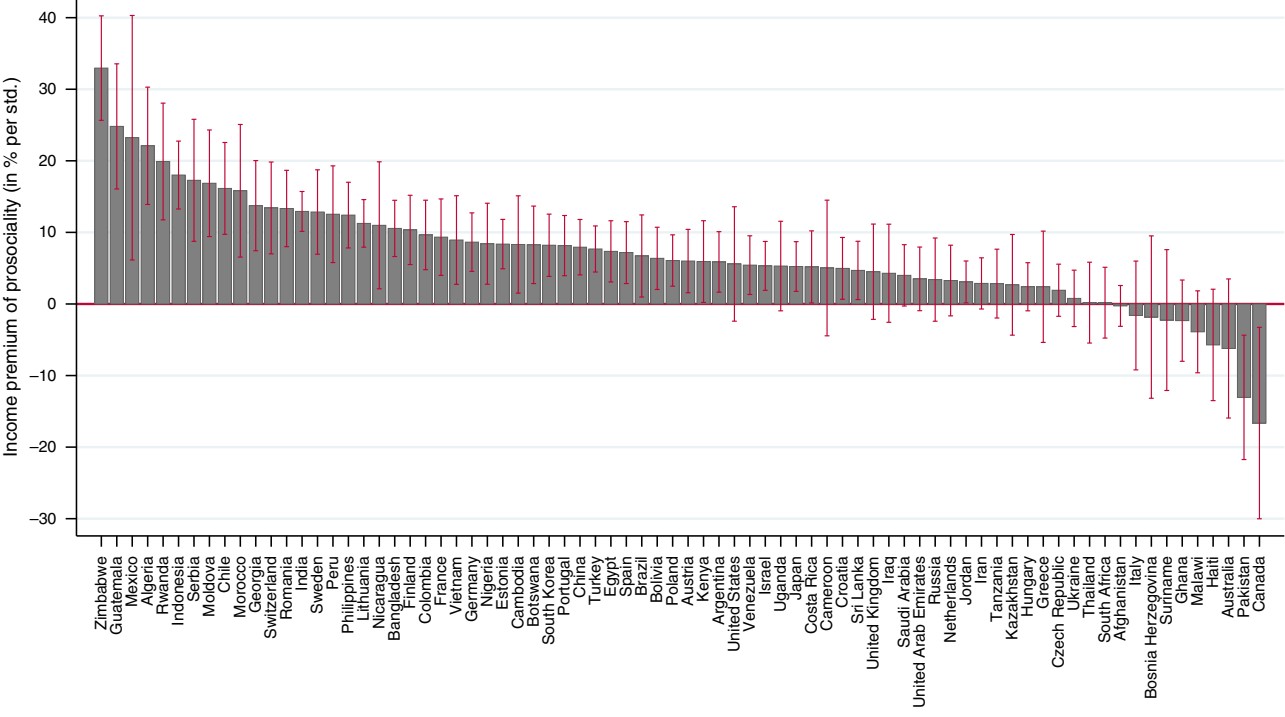

**Fig. 1 Income premia of prosociality (in percent of household income) around the world.** Displayed coefficients are country-specific estimates of the model shown in Table 1, Panel A, column 2, error bars indicate 90% confidence intervals.

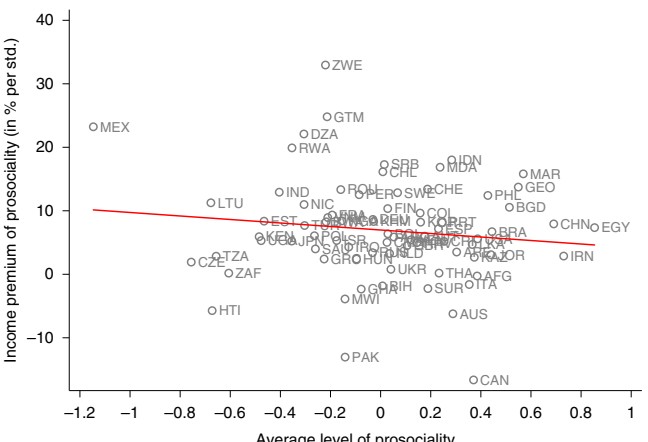

**Fig. 2 The relationship between income premia of prosociality and the average level of prosociality.** The red line indicates the prediction from a linear regression. The red line indicates the prediction from a linear regression. Spearman correlation: −0.091 ($p = 0.436$).

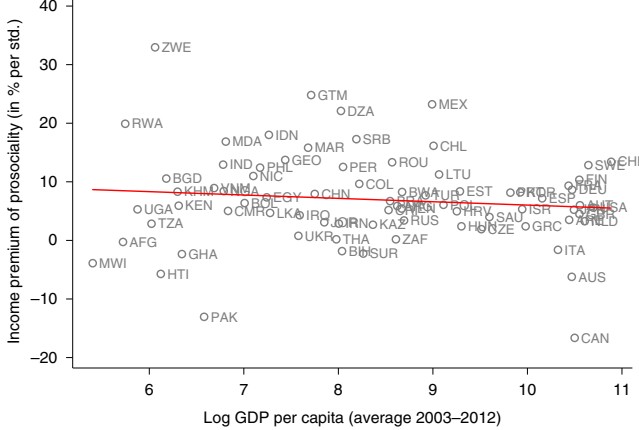

**Fig. 3 The relationship between income premia of prosociality and log GDP per capita.** The red line indicates the prediction from a linear regression. Spearman correlation: −0.065 ($p = 0.579$).

**Prosociality**. We refer to prosociality as positive other-regarding behaviors and beliefs. To yield a comprehensive measure of individuals' prosociality, we combine measures of three main facets: altruism, trust, and reciprocity (for a discussion see below and Kosse et al.[10]). Altruism reflects an individual's willingness to benefit others (without expecting anything in return), (positive) reciprocity reflects an individual's willingness to reward kind behavior, and trust indicates prosocial beliefs about the actions of others. The respective measures in the GPS combine information collected in form of hypothetical choice experiments and survey items. They were selected in an ex ante experimental validation procedure among large sets of items in order to exhibit the highest predictive power for corresponding incentivized behavioral measures. The final GPS measures are weighted scores of the respective items. Details regarding the measures and the experimental validation are described by Falk et al.[17,23]. We aggregate the three (standardized) facets of prosociality using principal component analysis (PCA), for details see the Methods section. In the analyses we use the resulting (standardized) first principal component as our measure of prosociality.

The survey questions in the GPS are comprised of a mixture of both qualitative quantitative items. Altruism was elicited by (i) the quantitative value in response to the question "Imagine the following situation: Today you unexpectedly received 1000 euros. How much of this amount would you donate to a good cause?" and (ii) the response on an 11-point Likert scale to the question "How willing are you to give to good causes without expecting anything in return?" Positive reciprocity was elicited by (i) an item asking for the value of a thank-you gift the respondent is willing to give in return for help by a stranger and (ii) the response on an 11-point Likert scale to the item "When someone does me a favor I am willing to return it." Trust was elicited by the response on an 11-point Likert scale to the item "I assume that people have only the best intentions."

Our approach on how to estimate prosociality is based on the following empirical and theoretical considerations. The literature suggests that different aspects of positive other-regarding behaviors and beliefs are positively correlated and have a common component. For example, Altmann et al.[24] show a strong positive interpersonal correlation between positive reciprocity and trust based on incentivized choice experiments. Within the GPS, Falk et al.[17] show positive relations among altruism, positive reciprocity, and trust at the individual and at the

country level. Therefore, in order to yield a comprehensive measure of individuals' prosociality that reflects positive behaviors towards and beliefs about others, we combine the GPS measures of the three facets of prosociality—altruism, trust, and positive reciprocity—into one measure. In our sample the average inter-item correlation of these three measures is 0.228, which is typically seen as a good level of internal consistency for broader higher order constructs[25]. To obtain the common factor among the three facets, we estimate the first principal component using PCA. The eigenvalues of the components are 1.486 (first component), 0.898 (second component), and 0.616 (third component). Therefore, the Kaiser criterion ("eigenvalues greater than one" rule) also suggests a one-dimensional structure of the concept.

Note that Falk et al.[17] show close to zero correlation between the three facets of prosociality with negative reciprocity. Similar results are found in Egloff et al.[26]. Dohmen et al.[27] discuss that positive and negative reciprocity might have different roots and tap into different emotional responses. Conducting a PCA using measures of the three facets (altruism, trust, and positive reciprocity) and negative reciprocity yields two eigenvalues bigger than one, which suggests that these four measures cannot adequately be captured by one component (Kaiser criterion). For these theoretical and empirical reasons, we do not consider negative reciprocity to be an element of prosociality.

The cross-sectional character of the GPS data does not allow us to directly explore the stability of our measure of prosociality in our sample. However, an analysis using the data of Kosse et al.[10] indicates a relatively high level of stability of prosociality already at elementary school-age. For an age-adopted measure of prosociality covering the same three facets mentioned above, we find a test-retest correlation (with 16 months in-between) of 0.449 (Spearman's $\rho$, $p < 0.01$, $N = 607$). This level of consistency is very similar to the level of consistency of personality traits in the same age range[28], which suggests a trait-like level of stability for prosociality. At the same time, Kosse et al.[10] show, based on a randomized controlled trial, that enriching the social environment persistently increases prosociality in elementary school children. Therefore, prosociality reflects relatively stable other-regarding behaviors and beliefs that can be influenced during sensitive periods[29,30].

**Reporting summary**. Further information on research design is available in the Nature Research Reporting Summary linked to this article.

## Data availability

The data that support the findings of this study can be downloaded under www.briq-institute.org/global-preferences (GPS data) and can be purchased at www.gallup.com (Gallup World Poll).

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

## Acknowledgements

We thank Armin Falk and Uwe Sunde for cooperation on the work with the GPS and the Gallup data. For comments and discussions, we are grateful to Katherina Kosse, Markus Paulus, and Chris Roth. We thank Anna Lane for outstanding research assistance. Financial support through the German Research Foundation through CRC TR 190 (project number 280092119) is gratefully acknowledged.

## Author contributions

F.K. and M.T. contributed to the study design. F.K. performed the data analysis. F.K. and M.T. drafted the manuscript and approved the final version of the manuscript for submission.

## Funding

## Competing interests

The authors declare no competing interests.
