## [Peer Review File · Nature Communications]

REVIEWER COMMENTS

Reviewer #1 (Remarks to the Author):

This paper provides strong evidence about the correlation between economic outcomes and prosociality across the world. The paper's evidence is very convincing and is a powerful use of the GPS data. This evidence is particularly important for studying the relationship between economic outcomes and personality, where there is generally more concern about reproducibility and generalizability of results.

I have one major question with the analysis. The paper normalizes pro-sociality within each country to be mean zero and standard deviation one. There is nothing inherently problematic with this, but it makes all results relative to other in their country. Given the GPS data has been used elsewhere to make cross-country comparisons, it would be interesting, and potentially important, to know if the results are similar when normalizing globally rather than within country.

In addition, I have a few smaller points:

- 1) The discussion about cognitive ability is currently a bit confusing and could be improved.
- 2) It may be good to make it clear that the measure of under-employment is subjective (their own beliefs on if they are underemployed), where one could imagine those beliefs are affected by their prosociality.
- 3) There is likely not room for this, but I would be interested to know what results look like when allowing for some non-linearity, such as regressing the outcomes on quartiles of pro-sociality.
- 4) There also may not be room for this, but I would be interested in knowing if some of the returns to relative prosociality reported in Figure 1 can be explained by differences in the average level in the country.

Reviewer #2 (Remarks to the Author):

Review of:

NCOMMS-20-14061-T

Prosociality predicts labor market success: Evidence from 76 countries

The study explores the association between prosociality and labor market success on a very large sample from 76 countries around the world. Its main finding is that there exist a positive association, which is robust across countries and continents, and after controlling for various demographic variables. Its main contribution appears to lie with the extension of known relations to non-WEIRD societies.

There is much to appreciate in this research given the sheer scale of the data, the apparent robustness of the associations, and the topic it deals with, which is of potential importance from a policymaking perspective.

However, in its present form there are some notable limitations.

Theory

1. The article makes causal arguments which are completely not grounded in the nature of the data. From reading the Title ("prosociality predicts") and Abstract (e.g., about experimentally valid measures of prosociality) this impression is further strengthened. However, it bears no hold in reality – the data is purely correlational.
2. The theoretical background is lacking, and basically non-existent. The reviewed literature is minimal, and it provides very little information about the state of knowledge on the focal question.
3. No attempt is made to explain the process that might account for the tested relations.

Method

4. Much of the relevant data concerning the present investigation is missing. Importantly, details about the measures involved. This leaves very little basis to independently evaluate the research. The reader is repeatedly referred to Falk et al. (2018), questioning the status of the present manuscript as independent research.
5. Explanation is not provided for the selection of the specific measures that comprised the prosociality index (i.e., why these are better than others? why has negative reciprocity not included?)
6. What exactly do the prosociality questions measure, a trait, a transient mental state? Given that the article suggests there is a basis for prosociality interventions, this issue is important.
7. Given that the prosociality measures are based on self-report (of prosocial tendencies), and not on behavioral indicators (e.g., money donated to charity causes), and certainly not on experimental intervention - the research needs to be more accurate in framing the effect as reflecting a relation of prosocial intentions with the DVs.
8. Not enough (actually, not at all) information is provided on the reliability of the composite index of prosociality.
9. Prosociality and interpersonal trust are not the same constructs (albeit they are related). It would be interesting to examine their individual contributions in the analyses.
10. Explanation is not provided for the selection of those particular control variables.

Results

11. The results do not present the direct effects associated with the controls (only the effects of the main predictor with/without the controls). I would expect more details on the correlations between all the measures involved, and about the independent contributions of the control variables.
12. Given that the main novelty is in the non-WEIRD countries: 1. Why should we expect different relations for non-WEIRD nations? (from a theoretical perspective) 2. Why aren't there any directed analyses comparing WEIRD and non-WEIRD nations?
13. Importantly, with no process to explain the findings and a structured model to test alternative explanations, the findings can be interpreted in multiple ways. There are many alternative explanations to the observed associations. For example, it makes sense to suggest that those who are in a better economic status (e.g., employed) are more likely to be prosocial than those in a bad economic status (unemployed), or that individuals in a low socioeconomic status are more egalitarian (as Piff, Kraus, Côté, Cheng, & Keltner, D., 2010, suggest). That is, the opposite causal direction from the one proposed in the article is as reasonably (or even more so) to explain the observed correlations.
14. The analysis indicates that nation's GDP does not moderate the effect. However, other factors that differentiate nations were not explored. The same is true for potential non-linear relations.

Discussion

15. Like the Introduction, the Discussion lacks in accounting for the theoretical (and even practical) contributions of the findings.
16. There is no discussion of the limitations of the study, and no suggestion for future research.

Responses to the comments of Reviewer #1

We would like to thank you for your positive reply and helpful suggestions on how to improve the paper. We were glad to hear that you think that “(t)he paper’s evidence is very convincing and is a powerful use of the GPS data”. In line with your comments, the comments of the other reviewer and the editor’s comments, we have substantially revised the paper. In particular, we now standardize prosociality within each country and discuss the role of cognitive ability and the subjective nature of underemployment in more detail. Moreover, we provide new analyses regarding non-linearities and between-country heterogeneity.

The revised version of the manuscript addresses all of your comments and suggestions. We believe that the changes made in response to your comments have markedly improved the paper. In the following, we explain how we have addressed your remarks.

Main Comment

The paper normalizes pro-sociality within each country to be mean zero and standard deviation one. There is nothing inherently problematic with this, but it makes all results relative to other in their country. Given the GPS data has been used elsewhere to make cross-country comparisons, it would be interesting, and potentially important, to know if the results are similar when normalizing globally rather than within country.

Thank you for pointing us to this link to the existing literature using the GPS dataset. In the revised version of the manuscript, we now thoroughly standardize prosociality using the global distribution to enable a direct comparison with the literature. The results based on the new standardization are displayed in the new Table 1 and indicate very similar but slightly stronger relations between labor market outcomes and prosociality: e.g. in the previous estimation a one standard deviation increase in prosociality predicted a 6.8% higher household income (without further controls, Panel A, column 1), in the new estimation a one standard deviation increase in prosociality predicts a 7.9% higher household income. A big advantage of the global-level standardization is, as you mentioned, that it allows for direct between country comparisons. We will make use of this when discussing your comment no. 4.

Further Comments

1. *The discussion about cognitive ability is currently a bit confusing and could be improved.*

We agree and therefore rewrote the discussion about cognitive ability. Particularly, we now discuss the conditions under which controlling for cognitive ability leads to an underestimation of the effect of interest.

“Controlling for cognitive ability moderately reduces the coefficient of prosociality (column 3). While controlling for cognitive ability is common in this literature, for an overview see Humphries and Kosse (2017), due to a bad control problem (Angrist and Pischke, 2008), the interpretation of this effect is ambiguous. Controlling for cognitive ability leads to an underestimation of the effect of interest if cognitive ability itself is a function of prosociality. This may be the case if both characteristics develop jointly and interactively (e.g., see Cunha and Heckmann (2007)). For a detailed discussion see Falk, Kosse, and Pinger (2020).”

2. *It may be good to make it clear that the measure of under-employment is subjective (their own beliefs on if they are underemployed), where one could imagine those beliefs are affected by their prosociality.*

We agree and added the following sentence to the discussion of the results:

“Note that the measure of underemployment is subjective as it relies on subjective beliefs on the appropriate capacity of work, which could directly be affected by prosociality.”

3. *There is likely not room for this, but I would be interested to know what results look like when allowing for some non-linearity, such as regressing the outcomes on quartiles of pro-sociality.*

We appreciate this suggestion and agree that such an analysis complements the linear approach. Therefore, building on your suggestion we added the following analyses to the revised version of the manuscript:

“In Table 2 we further check for potential within-country non-linear relations between labor market outcomes and prosociality by regressing labor market success on prosociality quartile dummies. In line with the previous analysis, the results indicate more labor market success for higher quartiles, which suggests positive marginal effects of prosociality for most parts of the distribution. The effects seem to be relatively stronger at the bottom and the middle parts of the distribution and relatively smaller at the top.”

	(1)	(2)	(3)
	Log HH income	Underemployed	Unemployed
Base: Prosociality 1st quarter			
Prosociality in 2nd quarter (dummy)	0.088*** (0.019)	-0.019* (0.010)	-0.011* (0.006)
Prosociality in 3rd quarter (dummy)	0.155*** (0.020)	-0.038*** (0.010)	-0.021** (0.008)
Prosociality in 4th quarter (dummy)	0.188*** (0.024)	-0.033*** (0.012)	-0.021** (0.008)
Observations	77,522	45,677	45,677

Table 2: Analyses of non-linear relations between labor market success and prosociality. Coefficients are OLS estimates, standard errors (clustered at country level) are displayed in parentheses, observations are weighted by the sampling weights provided by Gallup to achieve (ex post) representativeness. All regressions include sub-national region fixed effects. HH means household. Base category are individuals with prosociality in the bottom 25% of the global distribution. Prosociality in 2nd quarter is a dummy variable indicating whether an individual’s prosociality lies above the bottom 25% and below the median. Prosociality in 3rd quarter is a dummy variable indicating whether an individual’s prosociality lies above the median and below the top 25%. Prosociality in 4th quarter is a dummy variable indicating whether an individual’s prosociality lies in the top 25%. Data source: GPS and Gallup World Poll (76 countries). Significance levels regarding two-sided t-tests: * $p < 0.1$, ** $p < 0.05$, *** $p < 0.01$.

4. *There also may not be room for this, but I would be interested in knowing if some of the returns to relative prosociality reported in Figure 1 can be explained by differences in the average level in the country.*

We appreciate this suggestion and added the following analysis to the revised version of the manuscript:

“Moreover, we also explore the relation between countries’ income premia of prosociality, countries’ average level of prosociality and countries’ economic development. Figure 2 displays the relation between income premia and average level of prosociality and indicates no systematic relationship (Spearman’s $\rho = -0.091, p = 0.436, N = 76$). Figure 3 displays the relation between income premia and log GDP per capita and also indicates no systematic relationship (Spearman’s $\rho = -0.065, p = 0.579, N = 76$).”

Figure 2: The figure shows the relationship between income premia of prosociality and the average level of prosociality. The red line indicates the prediction from a linear regression.

Responses to the comments of Reviewer #2

We would like to thank you for your helpful comments and suggestions concerning how to improve the paper. We are glad that you think that “(t)here is much to appreciate in this research given the sheer scale of the data, the apparent robustness of the associations, and the topic it deals with, which is of potential importance from a policymaking perspective”. In line with your comments, the comments of the other reviewer and the editor’s comments, we have substantially revised the paper. We now provide clarifications about the correlational nature of the results, provide a discussion of the theoretical background and related processes, as well as clarifications about the methods and limitations of the study. Moreover, we provide new analyses regarding WEIRD and non-WEIRD countries and correlations among all control variables.

The revised version of the manuscript addresses all of your remarks. We believe that the changes made in response to your comments have substantially improved the paper.

Comments regarding Theory

1. *The article makes causal arguments which are completely not grounded in the nature of the data. From reading the Title (“prosociality predicts”) and Abstract (e.g., about experimentally valid measures of prosociality) this impression is further strengthened. However, it bears no hold in reality – the data is purely correlational.*

We fully agree that our manuscript only reports correlational results. In order to prevent misunderstandings regarding the interpretation of the reported correlations, we rewrote the manuscript and added clarifying statements. We now, e.g., state in the Introduction:

“We contribute to the literature by providing correlational evidence on the link between prosociality and labor market success in 76 representative country samples, which represent about 90% of the world population.”

According to our reading of the literature the term “predict” is commonly used to describe an empirical relation in the absence of a causal interpretation, e.g., Paul Allison writes in his textbook (Multiple Regression, 1999): “There are two main uses of multiple regression: prediction and causal analysis. In a prediction study, the goal is to develop a formula for making predictions about the dependent variable,

based on the observed values of the independent variable. (...) In a causal analysis, the independent variables are regarded as causes of the dependent variable.” Therefore, after consulting the editor, we decided to keep the title as is and clarify the correlational nature of our results in the main text as discussed above.

The expression “experimentally validated” is in line with the wording used by the authors of the GPS (Falk et al., 2018) and refers to the validation process of the survey measures: survey items included in the GPS data were selected in an ex ante experimental validation procedure. 402 subjects participated in incentivized-choice experiments and responded to a large set of survey items, which were either newly developed or taken from existing surveys. To construct the GPS, the authors selected those survey items that exhibit the highest predictive power for the corresponding incentivized behavioral measure. For clarification we now write:

“The respective measures in the GPS combine information collected in form of hypothetical choice experiments and survey items. They were selected in an ex ante experimental validation procedure among large sets of items in order to exhibit the highest predictive power for corresponding incentivized behavioral measures.”

2. *The theoretical background is lacking, and basically non-existent. The reviewed literature is minimal, and it provides very little information about the state of knowledge on the focal question.*
3. *No attempt is made to explain the process that might account for the tested relations.*

We followed your advice and extended the discussion of the related literature in the first section of the revised manuscript. We refer to additional related papers and especially mention some arguments from the literature on the related theoretical processes:

“Elements of prosociality such as reciprocity, altruism and trust are fundamental drivers of human interactions and affect a wide range of economic decisions. A large literature points to their importance for the functioning of societies and markets (Knack and Keefer, 1997; La Porta et al., 1997; Fehr and Gächter, 2002; Ostrom et al., 2002; Ashraf and Bandiera, 2017). While trust has been shown to be a driver of economic exchange (Guiso, Sapienza, and Zingales, 2009), especially reciprocity may work as contract enforcement

device and therefore lead to more efficient market outcomes (Fehr, Gächter, and Kirchsteiger, 1997). More recent evidence also points to the benefits of prosociality and social skills for individuals' life-outcomes (Becker et al., 2012; Deming, 2017; Kosse et al., 2020), which is in line with findings from a large body of literature on the returns to non-cognitive or socio-emotional skills (Heckman and Rubinstein, 2001; Heckman, Stixrud, and Urzua, 2006; Humphries and Kosse, 2017; Todd and Zhang, 2020). To describe the potential underlying processes, Deming (2017) presents a model which suggests that social skills reduce coordination costs, allowing workers to specialize and work together more efficiently. However, most of the evidence on the importance of prosociality is based on observations from Western, Educated, Industrialized, Rich, and Democratic (WEIRD) societies, which questions the generalizability of these results (Henrich, Heine, Norenzayan, 2010). Differences among societies might, e.g., evolutionarily arise from heterogeneities in market integration, community size and religious institutions, see Henrich et al. (2010) for a detailed theoretical and empirical discussion. ”

Our understanding of the literature on the predictive power of skills and personality measures is that, at the current stage, it is mostly an empirical debate about which measures predict success in which situations. We add to this literature by analyzing a sample which represent 90% of the world population instead of specific groups in specific countries. As the data at hand does unfortunately not allow us to further explore the underlying processes, we prefer to avoid further speculations about them and keep the discussion in a compact format.

Comments regarding Method

4. *Much of the relevant data concerning the present investigation is missing. Importantly, details about the measures involved. This leaves very little basis to independently evaluate the research. The reader is repeatedly referred to Falk et al. (2018), questioning the status of the present manuscript as independent research.*

Falk et al. (2018) describe the data collection process of the GPS and analyze differences in economics preferences among countries but do not explore determinants of labor market success. We follow your suggestion and extended the data description in the revised version of the manuscript and added a new Methods section that

addresses your comment. Especially, we now describe the items selection procedure when discussing our measure of prosociality and list the specific items in the Methods section:

“To yield a comprehensive measure of individuals’ prosociality we combine measures of three main facets: altruism, trust, and reciprocity (for a discussion see also Methods section and Kosse et al., 2020)). Altruism reflects an individual’s willingness to benefit others (without expecting anything in return), (positive) reciprocity reflects an individual’s willingness to reward kind behavior and trust indicates prosocial beliefs in the actions of others. The respective measures in the GPS combine information collected in form of hypothetical choice experiments and survey items. They were selected in an ex ante experimental validation procedure among large sets of items in order to exhibit the highest predictive power for corresponding incentivized behavioral measures.”

“The survey questions in the GPS comprised a mixture of qualitative items and quantitative items. Altruism was elicited by (i) the quantitative value in response to the question “Imagine the following situation: Today you unexpectedly received 1000 euros. How much of this amount would you donate to a good cause?” and (ii) the response on an 11-point Likert scale to the question “How willing are you to give to good causes without expecting anything in return?” Positive reciprocity was elicited by (i) an item asking for the value of a thank-you gift the respondent is willing to give in return for help by a stranger and (ii) the response on an 11-point Likert scale to the item “When someone does me a favor I am willing to return it.” Trust was elicited by the response on an 11-point Likert scale to the item “I assume that people have only the best intentions.” The final GPS measures are weighted scores of the respective items. Details regarding the measures and the experimental validation are described in Falk et al. (2016, 2018).”

5. *Explanation is not provided for the selection of the specific measures that comprised the prosociality index (i.e., why these are better than others? why has negative reciprocity not included?)*

In the revised version of the manuscript we now discuss details on the selection of measures in the new Methods section:

“Our approach on how to estimate prosociality is based on the following empirical and theoretical considerations. The literature suggests that different aspects of positive other-regarding behaviors and beliefs are positively correlated and have a common component. E.g., Altmann, Dohmen, and Wibral (2008) show based on incentivized choice experiments a strong positive interpersonal correlation between positive reciprocity and trust. Within the GPS, Falk et al (2018) show positive relations among altruism, positive reciprocity and trust at the individual and at the country level. Therefore, in order to yield a comprehensive measure of individuals’ prosociality that reflects positive behaviors towards and beliefs about others, we combine the GPS measures of the three facets of prosociality – altruism, trust, and positive reciprocity – into one measure. In our sample the average inter-item correlation of these three measures is 0.228, which is usually seen as a good level of internal consistency for broader higher order constructs (Clark and Watson, 1995). To obtain the common factor among the three facets we estimate the first principal component using PCA. The eigenvalues of the components are 1.486 (first component), 0.898 (second component) and 0.616 (third component). Therefore, the Kaiser criterion (“eigenvalues greater than one” rule) also suggests a one-dimensional structure of the concept.

Note that Falk et al. (2018) show close to zero correlations of the three facets of prosociality with negative reciprocity. Similar results are found in Egloff, Richter, and Schmukle (2013). Dohmen et al. (2008) discuss that positive and negative reciprocity might have different roots and tap into different emotional responses. Conducting a PCA using measures of the three facets (altruism, trust, and positive reciprocity) and negative reciprocity yields two eigenvalues bigger than one, which suggests that these four measures cannot adequately be captured by one component (Kaiser criterion). For these theoretical and empirical reasons we do not consider negative reciprocity as part of prosociality.”

6. *What exactly do the prosociality questions measure, a trait, a transient mental state? Given that the article suggests there is a basis for prosociality interventions, this issue is important.*

Next to the above mentioned discussion in the Methods section, we now clarify our understanding of the concept of prosociality also in the main text:

“Prosociality reflects positive behaviors towards and beliefs about others.”

Given the cross-sectional character of the GPS data, we are unfortunately not able to directly explore the stability of our prosociality measure within our sample. However, we added a discussion on this issue and present evidence from another sample in the Methods section.

“The cross-sectional character of the GPS data does not allow us to directly explore the stability of our measure of prosociality in our sample. However, an analysis using the data of Kosse et al. (2020) indicates a relatively high level of stability of prosociality already at elementary school-age. For an age-adopted measure of prosociality, which covers the same three facets as mentioned above, we find a test-retest correlation (with 16 months in-between) of 0.449 (Spearman’s $\rho, p < 0.01, N = 607$). This level of consistency is very similar to the level of consistency of personality traits in the same age range (Roberts and DelVecchio, 2000), which suggests a trait-like level of stability for prosociality.”

7. *Given that the prosociality measures are based on self-report (of prosocial tendencies), and not on behavioral indicators (e.g., money donated to charity causes), and certainly not on experimental intervention - the research needs to be more accurate in framing the effect as reflecting a relation of prosocial intentions with the DVs.*

As described in our previous answers, we now clarify our understanding of the concept of prosociality as “positive behaviors towards and beliefs about others” and describe the items in detail. Moreover, we point to the fact that the items are chosen to highly correlate with incentivized behavioral measures.

“The respective measures in the GPS combine information collected in form of hypothetical choice experiments and survey items. They were selected in an ex ante experimental validation procedure among large sets of items in order to exhibit the highest predictive power for corresponding incentivized behavioral measures.”

8. *Not enough (actually, not at all) information is provided on the reliability of the composite index of prosociality.*

In the new Methods section we now discuss internal consistency and the Kaiser criterion.

“In our sample the average inter-item correlation of these three measures is 0.228 which is usually seen as a good level of internal consistency for broader higher order constructs (Clark and Watson, 1995). To obtain the common factor among the three facets we estimate the first principal component using PCA. The eigenvalue of the first component is 1.486, the eigenvalue of the second component is 0.898 and the eigenvalue of the third component is 0.616. Therefore, the Kaiser criterion (“eigenvalues greater than one” rule) also suggest a one-dimensional structure of the concept.”

9. *Prosociality and interpersonal trust are not the same constructs (albeit they are related). It would be interesting to examine their individual contributions in the analyses.*

We appreciate this suggestion and now also explore the predictive power of the facets separately.

“In Table 3 we explore the predictive power of the individual facets altruism, positive reciprocity and trust. The estimates indicate that all three facets individually predict labor market success. with positive reciprocity showing the highest and trust the lowest wage premium.”

	(1)	(2)	(3)
	Log household income		
Altruism (standardized)	0.057*** (0.008)		
Positive reciprocity (standardized)		0.074*** (0.007)	
Trust (standardized)			0.016** (0.007)
Observations	77,522	77,522	77,522

Table 3: Altruism, reciprocity and trust predict labor market success. Coefficients are OLS estimates, standard errors (clustered at country level) are displayed in parentheses, observations are weighted by the sampling weights provided by Gallup to achieve (ex post) representativeness. All regressions include sub-national region fixed effects and controls for age and gender (see Table 1, column 2). Data source: GPS and Gallup World Poll (76 countries). Significance levels regarding two-sided t-tests: * $p < 0.1$, ** $p < 0.05$, *** $p < 0.01$.

10. *Explanation is not provided for the selection of those particular control variables.*

In the revised version of the manuscript we extended the discussion regarding controlling for socio-demographics and IQ.

“The results, shown in Table 1 Panel A, indicate that prosociality positively predicts household income. Without further controls, an increase of one standard deviation in prosociality predicts an about 8% higher household income. As labor market outcomes and prosociality might both be determined by socio-demographics, we control for gender and age in column 2 which, however, does not affect the relation between prosociality and earnings (column 2). Controlling for cognitive ability moderately reduces the coefficient of prosociality (column 3). While controlling for cognitive ability is a common in this literature, for an overview see Humphries and Kosse (2017), due to a bad control problem (Angrist and Pischke, 2008), the interpretation of this effect is ambiguous. Controlling for cognitive ability leads to an underestimation of the effect of interest if cognitive ability itself is a function of prosociality. This may be the case if both characteristics develop jointly and interactively (e.g., see Cunha and Heckmann (2007)). For a detailed discussion see Falk, Kosse, and Pinger (2020).”

Comments regarding Results

11. *The results do not present the direct effects associated with the controls (only the effects of the main predictor with/without the controls). I would expect more details on the correlations between all the measures involved, and about the independent contributions of the control variables.*

As discussed in our previous answer, we follow your advice and we now discuss the role of the used control variables in more details. Moreover, we show the requested analyses in Tables S1 and S2. Table S1 shows all coefficients estimated in Table 1, column 3. The results are in line with the previous literature: Individuals with more cognitive ability have better labor market outcomes. There is a concave relation between age and labor market outcomes and females earn less and are more often under- or unemployed. Table S2 shows correlations between all the measures involved and confirms the patterns in the regression results.

	(1)	(2)	(3)
	Log HH income	Underemployed	Unemployed
Prosociality (standardized)	0.060*** (0.009)	-0.011*** (0.004)	-0.008** (0.003)
Cognitive ability (std.)	0.101*** (0.008)	-0.008* (0.004)	-0.003 (0.003)
Age (in years)	0.011*** (0.002)	-0.017*** (0.001)	-0.014*** (0.001)
Age squared / 100	-0.018*** (0.002)	0.015*** (0.001)	0.013*** (0.001)
Female dummy	-0.060*** (0.013)	0.064*** (0.009)	0.048*** (0.009)
Observations	77,522	45,677	45,677

Table S1: Coefficients are OLS estimates, standard errors (clustered at country level) are displayed in parentheses, observations are weighted by the sampling weights provided by Gallup to achieve (ex post) representativeness. All regressions include sub-national region fixed effects. HH means household. Data source: GPS and Gallup World Poll (76 countries). Significance levels regarding two-sided t-tests: * $p < 0.1$, ** $p < 0.05$, *** $p < 0.01$.

Variables	(1)	(2)	(3)	(4)	(5)	(6)	(7)
(1) Log HH income	1						
(2) Underemployed	-0.187	1.000					
(3) Unemployed	-0.127	0.688	1.000				
(4) Prosociality	0.089	-0.042	-0.032	1.000			
(5) Cognitive ability	0.218	-0.048	-0.024	0.177	1.000		
(6) Age (in years)	0.138	-0.175	-0.172	0.029	-0.052	1.000	
(7) Female dummy	-0.054	0.073	0.069	0.035	-0.011	0.005	1.000

Table S2: Individual level pairwise Spearman correlation coefficients for all used variables. HH means household. Observations between 45,686 and 77,536. Data source: GPS and Gallup World Poll (76 countries).

12. *Given that the main novelty is in the non-WEIRD countries: 1. Why should we expect different relations for non-WEIRD nations? (from a theoretical perspective) 2. Why aren't there any directed analyses comparing WEIRD and non-WEIRD nations?*

We believe that the main contribution of the paper is the empirical analysis of the relation between prosociality and labor market outcomes in a sample that represents

90% of the world population. To build on your suggestion we extend the discussion in the introduction and mainly refer to the literature regarding further theoretical considerations. Moreover, we added an empirical comparison to the analysis.

“Differences among societies might, e.g., evolutionarily arise from heterogeneities in market integration, community size and religious institutions, see Henrich et al. (2010) for a detailed theoretical and empirical discussion.”

“To explore the heterogeneity among country-specific income premia, we further regress the estimated income premia on continent dummies. A Wald test does not reject the null hypothesis that all dummies are jointly zero ($p = 0.520, N = 76$) which indicates that income premia do not significantly differ across continents. Similarly, we also find no statistically significant difference in income premia between WEIRD countries, proxied by North America, Europe and Australia, and non-WEIRD countries, proxied by Asia, Africa and South America ($p = 0.711$, two-sided t-test, $N = 76$). Moreover, we also explore the relation between countries’ income premia of prosociality, countries’ average level of prosociality and countries’ economic development. Figure 2 displays the relation between income premia and average level of prosociality and indicates no systematic relationship (Spearman’s $\rho = -0.091, p = 0.436, N = 76$). Figure 3 displays the relation between income premia and log GDP per capita and also indicates no systematic relationship (Spearman’s $\rho = -0.065, p = 0.579, N = 76$).”

13. *Importantly, with no process to explain the findings and a structured model to test alternative explanations, the findings can be interpreted in multiple ways. There are many alternative explanations to the observed associations. For example, it makes sense to suggest that those who are in a better economic status (e.g., employed) are more likely to be prosocial than those in a bad economic status (unemployed), or that individuals in a low socioeconomic status are more egalitarian (as Piff, Kraus, Côté, Cheng, & Keltner, D., 2010, suggest). That is, the opposite causal direction from the one proposed in the article is as reasonably (or even more so) to explain the observed correlations.*

As discussed in our answer to your first comment, we are fully aware of the correlational nature of our results and rewrote several parts of the manuscript to make this clear. Like most other related studies, ours has no ambition to tease out causality. E.g. Tables 2 and 3 in Kosse and Humphries (2017) list 29 empirical papers on the

relation of different measures of non-cognitive skills and life outcomes. The vast majority of them reports correlations. The few studies which try to establish causal relations rely on very data demanding methods and strict assumptions (e.g., Heckman et al., 2006). Therefore, our study is in line with most of the existing literature regarding analysis, interpretation and wording. We contribute to the literature by providing a global perspective.

14. *The analysis indicates that nation's GDP does not moderate the effect. However, other factors that differentiate nations were not explored. The same is true for potential non-linear relations.*

We have no ambition to provide a complete analysis of heterogeneity across countries. On the contrary, we see this as a starting point of future research, which we now clarify in the discussion (see below).

We added a second analysis of heterogeneity across countries and explore the relation between the average level of prosociality and the income premium of prosociality in the new Figure 2. As for the relation of GDP and income premium, we do not find a statistically significant rank correlation and the scatter plots do also not suggest systematic nonlinear relations.

Comments regarding Discussion

15. *Like the Introduction, the Discussion lacks in accounting for the theoretical (and even practical) contributions of the findings.*

We appreciate this suggestion and now explicitly point to the theoretical and practical contributions of our findings.

“Income premia of prosociality are not systematically different across continents and are not related to countries’ average level of prosociality or countries’ economic development. This further underlines the generality of the found relation and suggests that novel models of labor markets that include prosociality and social skills (as e.g., Deming, 2017) are likely to have a broad scope.”

16. *There is no discussion of the limitations of the study, and no suggestion for future research.*

We follow your suggestion and now explicitly point to the exploration of between-country heterogeneity as important future research in the discussion.

“(...) our findings ask for future research on the determinants of the between-country heterogeneity.”

Moreover, we decided to refer to the limitations of the study as soon as they occur in the text. We write, e.g., in the data description “(t)he Gallup World Poll does not include information on individuals’ earnings but provides three internationally comparable measures of labor market success (...)” and in the Methods section “(t)he cross-sectional character of the GPS data does not allow us to directly explore the stability of our measure of prosociality in our sample.”

REVIEWER COMMENTS

Reviewer #1 (Remarks to the Author):

The revisions in response to my comments and the comments to the other reviewer have notably improved the paper. All of my previous concerns have been addressed. Reading the new version, I have two small comments:

1. The paper would benefit from some copy editing.
2. When discussing Table 2, the paper could reach somewhat stronger conclusions that moving from the 3rd to 4th decile having smaller effects than moving from the 1st to 2nd or 2nd to 3rd.

Reviewer #2 (Remarks to the Author):

Review of:

NCOMMS-20-14061A

Prosociality predicts labor market success: Evidence from 76 countries

The authors have generally been responsive to the comments.

I still find myself unsatisfied with the overall conclusion of the research, given the vast alternative explanations of these correlations, and the absence of evidence to rule-out alternative explanations (the basic ones would be that there is no causal relation between the two variables, or that the causal relation is reversed, that is, that wealthy people are more prosocial.). I understand that given the present data, no empirical response can be provided to address this limitation, but I would expect a more direct discussion of this issue as a major limitation. The fact that a relation between two variables is documented across the globe (a point repeatedly mentioned as the Study's main contribution) does not make it any more informative about the nature of the association and its causes.

In addition, the nature of prosociality is still not explained well enough. Specifically, it is not clear whether the authors refer to it as a behavioral tendency (as the Introduction reads "prosociality reflects positive behaviors") or as a stable trait (as indicated in the Methods section " This level of consistency is very similar to the level of consistency of personality traits"). This question is important in discussing the implications of the research (assuming causal relations between prosociality and success). Behaviors can be modified, whereas traits are generally stable, and the practical implications are vastly different. The final sentence in the Discussion about the malleability of prosociality is thus incompatible with the description of this construct as a trait-like in the Method section.

Responses to the comments of Reviewer #1

We would like to thank you for your positive reply and helpful suggestions on how to improve the paper. We were glad to hear that you think that “(t)he revisions in response to my comments and the comments to the other reviewer have notably improved the paper”. In line with your comments, the comments of the other reviewer and the editor’s comments, we have revised the paper. In particular, we did careful copy-editing and adapted the discussion regarding Table 2. In the following, we explain how we have addressed your remarks.

Comments

1. *The paper would benefit from some copy editing.*

We agree and worked together with a copy-editor to improve the manuscript.

2. *When discussing Table 2, the paper could reach somewhat stronger conclusions that moving from the 3rd to 4th decile having smaller effects than moving from the 1st to 2nd or 2nd to 3rd.*

We agree and now explicitly refer to the concavity of the relation between labor market success and prosociality

“... the results indicate more labor market success for higher quartiles and suggest positive marginal effects of prosociality for most parts of the distribution. However, the effects are relatively stronger at the bottom and middle parts of the distribution and relatively smaller at the top, which suggests a concave relationship between labor market success and prosociality.”

Responses to the comments of Reviewer #2

We would like to thank you for your helpful comments and suggestions concerning how to improve the paper. We are glad that you think that “(t)he authors have generally been responsive to the comments”. In line with your comments, the comments of the other reviewer and the editor’s comments, we have revised the paper. In particular, we discuss the correlational nature of the findings in the conclusion section and the properties of our measure of prosociality in the methods section. We believe that the changes made in response to your comments have substantially improved the paper.

Comments

1. *I still find myself unsatisfied with the overall conclusion of the research, given the vast alternative explanations of these correlations, and the absence of evidence to rule-out alternative explanations (the basic ones would be that there is no causal relation between the two variables, or that the causal relation is reversed, that is, that wealthy people are more prosocial.). I understand that given the present data, no empirical response can be provided to address this limitation, but I would expect a more direct discussion of this issue as a major limitation. The fact that a relation between two variables is documented across the globe (a point repeatedly mentioned as the Study’s main contribution) does not make it any more informative about the nature of the association and its causes.*

Following your suggestion, we refer to the correlational nature of our findings in the Introduction and the Analysis sections, and we also added a more direct discussion in the Conclusion section.

Introduction: “We contribute to the literature by providing correlational evidence on the link between prosociality and labor market success in 76 representative country samples (...).”

Analysis: “We label this link, which is not necessarily causal, as income premium of prosociality.”

Conclusion: “We contribute to the debate regarding generalizability of results in the social sciences (Henrich, Heine, and Norenzayan, 2010) by showing that around the globe, prosociality is generally positively related to labor market success. The finding robustly holds in a pooled sample which represents about

90% of the world population and the pattern also holds for the majority of individual countries. Like most other related studies (for an overview see, e.g., Humphries and Kosse (2017)), ours has no ambition to tease out causality. The cross-sectional character of our data does not allow us to explore reverse causality or omitted variables.”

2. *In addition, the nature of prosociality is still not explained well enough. Specifically, it is not clear whether the authors refer to it as a behavioral tendency (as the Introduction reads "prosociality reflects positive behaviors") or as a stable trait (as indicated in the Methods section "This level of consistency is very similar to the level of consistency of personality traits"). This question is important in discussing the implications of the research (assuming causal relations between prosociality and success). Behaviors can be modified, whereas traits are generally stable, and the practical implications are vastly different. The final sentence in the Discussion about the malleability of prosociality is thus incompatible with the description of this construct as a trait-like in the Method section.*

In order to prevent misunderstandings across disciplines we abstain from using discipline-specific labels. Instead, we discuss the properties of the construct in more detail in the revised version of the methods section.

“(…) an analysis using the data of Kosse et al. (2020) indicates a relatively high level of stability of prosociality already at elementary school-age. For an age-adapted measure of prosociality covering the same three facets mentioned above, we find a test-retest correlation (with 16 months in-between) of 0.449 (Spearman’s ρ , $p < 0.01$, $N = 607$). This level of consistency is very similar to the level of consistency of personality traits in the same age range (Roberts and DelVecchio, 2000), which suggests a trait-like level of stability for prosociality. At the same time, Kosse et al. (2020) show, based on a randomized controlled trial, that enriching the social environment persistently increases prosociality in elementary school children. Therefore, prosociality reflects relatively stable other-regarding behaviors and beliefs that can be influenced during sensitive periods. For further discussions, see also Knafo-Noam et al. (2015) and Paulus (2018).”